# FlexIR: Towards Flexible and Manipulable Image Restoration

## ABSTRACT

The domain of image restoration encompasses a wide array of highly effective models (*e.g.*, SwinIR, CODE, DnCNN), each exhibiting distinct advantages in either efficiency or performance. Selecting and deploying these models necessitate careful consideration of resource limitations. While some studies have explored dynamic restoration through the integration of an auxiliary network within a unified framework, these approaches often fall short in practical applications due to the complexities involved in training, retraining, and hyperparameter adjustment, as well as limitations as being totally controlled by auxiliary network and biased by training data. To address these challenges, we introduce FlexIR: a flexible and manipulable framework for image restoration. FlexIR is distinguished by three components: a meticulously designed hierarchical branch network enabling dynamic output, an innovative progressive self-distillation process, and a channel-wise evaluation method to enhance knowledge distillation efficiency. Additionally, we propose two novel inference methodologies to fully leverage FlexIR, catering to diverse user needs and deployment contexts. Through this framework, FlexIR achieves unparalleled performance across all branches, allowing users to navigate the trade-offs between quality, cost, and efficiency during the inference phase. Crucially, FlexIR employs a dynamic mechanism powered by a non-learning metric independent of training data, ensuring that FlexIR is entirely under the direct control of the user. Comprehensive experimental evaluations validate FlexIR's flexibility, manipulability, and cost-effectiveness, showcasing its potential for straightforward adjustments and quick adaptations across a range of scenarios. Codes will be available at [URL].

## 1 INTRODUCTION

Image restoration, a longstanding challenge, seeks to recover pristine images from degraded counterparts, has been applied in a wide range of industrial fields [16, 29, 44, 45]. The vast computational resources available and large amount of data have driven researchers to build powerful image restoration models with strong performance. Convolutional Neural Networks (CNNs) have dominantly addressed restoration tasks [20, 22, 41, 47, 56], yet their efficacy is curtailed by the basic convolution layer's limitations in capturing long-range dependencies. With the proposal of self-attention mechanism, transformer-based models have yielded impressive results [5, 21, 37, 46, 57] and become preferred choice for restoration problems.

*ACM MM, 2024, Melbourne, Australia*
© 2024 Copyright held by the owner/author(s). Publication rights licensed to ACM.
ACM ISBN 978-x-xxxx-xxxx-x/YY/MM
https://doi.org/10.1145/nnnnnnn.nnnnnnn

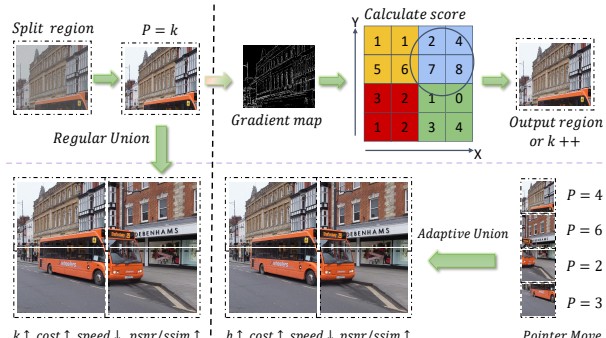

**Figure 1: Regular Branch Inference (left) and Adaptive Inference (right). In Regular Branch Inference, image regions are recovered and integrated by setting pointer $P$. For Adaptive Inference, the pointer $P$ is initially set to 1. Utilizing a gradient map, a score independent of the dataset is calculated for each recovered region. Here, $h$ denotes the threshold, enabling regions with scores exceeding this threshold to be promptly outputted. If the score does not surpass the threshold, the pointer activates the next B-RSTB ($k++$), optimizing the inference process for enhanced performance.**

Despite the abundance of models available for image restoration tasks, the current landscape predominantly offers models that cater to a coarse level of complexity and performance, each designed for a specific resource allocation (*e.g.*, FLOPs, GPU memory cost). Although certain models [21, 51, 57] exhibit remarkable capabilities, showcasing efficiency or performance orientation, they are inherently static and tailored, lacking the flexibility to adapt to varying resource limitations. In practice, the deployment of models across diverse platforms necessitates adaptability to distinct resource constraints, such as energy, latency, and on-chip memory. Recent efforts in developing dynamic restoration networks [18, 36, 43] aim to address these challenges by enabling automatic trade-offs. However, these solutions fall short in scalable real-world deployment. Their telepathic designs are employing an auxiliary network to aid unified networks in identifying patch-level difficulties, facilitating adaptive outputs post-training to minimize computational expenses. Yet, this methodology demands intricate training processes, and the output heavily depends on the model's predetermined behavior. Moreover, the auxiliary network's decision-making process is significantly influenced by the training dataset, posing generalization challenges in practical scenarios, which often require extensive retraining and hyperparameter tuning. A promising direction is the exploration of non-learning metrics that do not depend on training data [25], proposing a framework that allows human oversight over model behavior and resource consumption. In summary, while dynamic frameworks offer potential, their practical application remains limited by these constraints.

In this work, we introduce FlexIR, a flexible and manipulable framework designed for image restoration tasks. FlexIR maintains

the capability for dynamic outputs while eliminating the need for joint training with an auxiliary network. This approach renders the framework fully adaptable by users and suitable for a broad spectrum of deployment scenarios. At the core of FlexIR is a hierarchical network architecture, enhanced by our newly proposed Branch Residual Swin Transformer Blocks (B-RSTB), which supports dynamic output generation. This branched network design allows for straightforward manipulation during the inference stage to meet varying resource requirements, eliminating the need for retraining. To efficiently train the network, we devise a novel progressive self-distillation mechanism. This mechanism leverages a pointer system to sequentially activate B-RSTBs, aiming for theoretically optimal restoration quality at each step. Our investigations reveal that the knowledge (features) derived from the preceding B-RSTB affects subsequent blocks in unique ways. This insight has led us to introduce a channel-wise scoring mechanism for feature distillation, enhancing the training process. We detail two technical approaches for model inference: *1. Regular Branch Inference* and *2. Adaptive Inference* wherein a dataset-irrelevant and gradient-based scoring method is introduced. These approaches enable FlexIR to cater to diverse application needs, as illustrated in Fig. 1 and will be further discussed in methodology section. The primary contributions of our paper are summarized as follows:

- We introduce FlexIR, a Flexible and Manipulable Image Restoration Framework which stands out as a versatile, cost-effective solution tailored to meet varying user demands, capable of seamless deployment across a broad spectrum of platforms with distinct resource limitations.
- We present a novel progressive self-distillation mechanism alongside a channel-wise scoring mechanism to enhance model training, fostering greater efficiency and improved convergence results. Additionally, we detail two innovative approaches for model inference, incorporating adaptive inference facilitated by a dataset-independent scoring method. Two approaches amplifies FlexIR's utility across diverse application scenarios.
- Comprehensive experimental validations demonstrate our method's superiority. Specifically, the fully-equipped FlexIR model rivals state-of-the-art alternatives, while its compact variant outperforms other small-scale methods. Significantly, FlexIR's operational parameters during inference are manually adjustable, negating the need for complex retraining or adjustment of auxiliary networks characteristic of existing dynamic frameworks. This attribute underscores FlexIR's enhanced adaptability and capability to negotiate quality-cost-efficiency trade-offs effectively.

## 2 RELATED WORK

### 2.1 Image Restoration and Vision Transformer

Image restoration has long been dominated by CNN-based models, demonstrating impressive performance in various tasks such as image super-resolution [9], image denoising [51], and reducing JPEG compression artifacts [8]. These CNN models excel at local feature extraction and efficiently learn mappings between low-quality and high-quality images using large image pairs.

Inspired by the remarkable success of transformers in NLP field, researchers have ventured into applying transformer-based models to computer vision tasks, including image classification [10, 26, 31, 40], object detection [3, 23], and segmentation [40, 58]. In the domain of restoration tasks, recent methods have focused on achieving a better trade-off between speed and quality. For instance, Uformer [37] introduced a U-shape transformer-based structure with a window transformer block for image restoration. Restormer [46] adopted a U-shape structure and replaced the original spatial attention with channel attention to reduce computational intensity while performing attention in a lower dimension. SwinIR [21] proposed RSTB (Residual Swin Transformer Block) based on Swin Transformer [26], incorporating a window shift mechanism to reduce complexity and improve efficiency. More recently, Zhao [57] presented an efficient transformer model for image restoration, transferring feature aggregation at the pixel level into a lower-dimensional space of superpixels to avoid computationally expensive global self-attention.

Some of these methods prioritize efficiency, while others focus on effectiveness, nevertheless, they still fall under the category of fixed models and are inflexible at the inference stage and cannot adapt to diverse and dynamic deployment environment.

### 2.2 Dynamic, Flexible and Manipulable

The concept of dynamic neural networks seeks to enhance model flexibility by adapting processing pathways based on the varying complexity of input samples. This paradigm shift toward adaptive inference has spurred innovative approaches, for instance, Branchynet [33] introduced an early exiting strategy for image classification, allowing the model to exit from intermediate layers once it becomes confident enough in its predictions. Similarly, PABEE [59] proposed a Patience-based mechanism, demonstrating the feasibility of enhancing the efficiency of BERT [7] with theoretical analysis. FastBERT [24] further advanced this idea by combining self-distillation with a sample-wise adaptive mechanism, striking a balance between speed and accuracy in response to varying request amounts. MSDNet [14] and its variants [19, 42] develop a multi-classifier architecture for the image classification task. These methodologies, however, predominantly cater to classification problems and do not straightforwardly extend to the domain of image restoration.

| Feature\Method | FlexIR | Dynamic Methods [18, 36, 43] | Fixed Methods [21, 49, 51] |
|---|---|---|---|
| No Joint Training (No Auxiliary Net) | ✓ | ✗ | ✓ |
| Dynamic | ✓ (Optional) | ✓ | ✗ |
| Flexible and Manipulable | ✓ | ✗ | ✗ |
| Inference | User-Controlled Dataset-Irrelevant | Model-Determined Dataset-Trapped | Computation Cost Fixed |

In the realm of dynamic image restoration, the conventional approach employs an auxiliary network to gauge task difficulty at a granular level. Classsr [18] introduces a dynamic super-resolution strategy, leveraging a Class-Module for difficulty-based sub-image

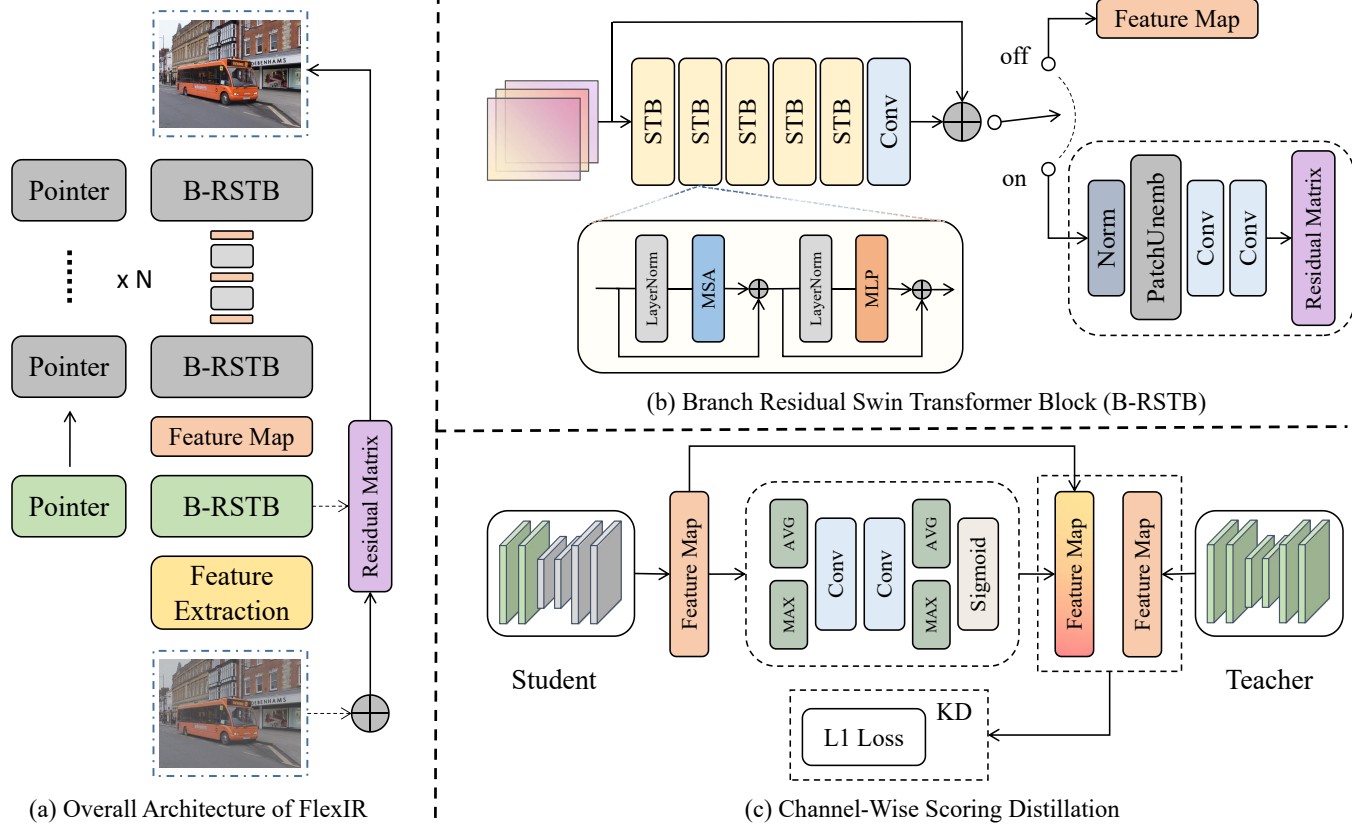

**Figure 2: Overview of FlexIR. (a) is the overall architecture of FlexIR, a pointer is utilized to activate B-RSTB step by step, which also enables progressive self-distillation in training stage. (b) is network structure of Branch Residual Swin Transformer Block (B-RSTB). A switch is associated with a pointer to determine whether to enter the branch network. (c) is the channel-wise scoring distillation process. Partial activated network serves as student and full activated network serve as teacher, while a score map $\phi$ is computed through a learnable channel attention network and student feature map is channel weighted for further feature distillation.**

classification followed by an SR-Module for resolution enhancement. Adaptive Patch Exiting [36] offers a scalable solution for super-resolution, employing a regressor to evaluate the incremental utility of each layer for a given patch. Path-Restore [43] pioneers the use of reinforcement learning in training a pathfinder to navigate the optimal processing route for each image, informed by a difficulty-regulated reward mechanism. Despite their innovations, these methods rely on auxiliary networks for functionality, necessitating joint training and model adjustments to refine output behaviors—a process that is both complex and data-dependent, challenging the generalizability [25] of these models in real-world applications.

Contrasting these approaches, our FlexIR framework is distinguished by its independence from auxiliary network-driven adjustments. FlexIR's operational parameters can be dynamically tailored at the inference stage, obviating the need for joint training or complex re-adjustment protocols. This design philosophy not only simplifies the application of FlexIR but also enhances its flexibility and manipulability to navigate quality-cost-efficiency trade-offs more effectively.

## 3 METHOD

### 3.1 Architecture

As shown in Fig. 2 (a), our network is the hierarchical architecture stacked with several Branch Residual Swin Transformer Blocks (B-RSTBs). In general inference manifold, given a degraded image $I_{LQ}$, we initially employ a standard feature extraction module to generate a shallow feature representation $e$. Subsequently, we feed the feature representation $e$ into first B-RSTB if is activated. As depicted in Fig. 2 (b), a pivotal *switch* is associated with a pointer, determining whether to enter the branch network. If the next B-RSTB is not activated by pointer, residual matrix $\mathcal{M}$ will be generated and added to original image to obtain restored image $\hat{I}_{HQ}$, Conversely, when the switch is off, the B-RSTB acts as an encoder and generates a new feature map, as indicated in Equation (1):

$$Output = \begin{cases} \hat{I}_{HQ} = \mathcal{M} + I_{LQ}, & switch \rightarrow on \\ F : \text{Feature Map}, & switch \rightarrow off \end{cases} \quad (1)$$

By progressively activating more B-RSTBs, our model enhances its encoder capability, thereby improving the restoration performance. However, it's essential to note that this also results in increased

memory and computing costs, but can be efficiently manipulated by adjusting the pointer settings.

## 3.2 Model Training

The objective of FlexIR training is to achieve optimal restoration results across all B-RSTBs. Differing from inference manifold, our training process necessitates the utilization of both the feature map $F$ and $\hat{I}_{HQ}$ for parameter optimization. One simple approach [24] if applied here is to direct activate all B-RSTBs and perform back-propagation collectively, this method may introduce optimization conflicts, particularly between deeper and shallower B-RSTBs. To mitigate this, we propose a progressive self-distillation approach for model training.

*3.2.1 **Progressive Self-Distillation**.* The process of progressive self-distillation (PSD) is associated with the movement of pointer. Following a predefined training schedule, pointer progressively activates B-RSTB step by step, while non-activated B-RSTBs do not output residual matrix and acquire gradient for optimization. Additionally, we introduce a teacher model, which is fully activated FlexIR directly loading the pre-trained parameters from SwinIR [21], to generate soft labels for feature distillation. **Algorithm 1** outlines the details of this process, including the loss strategy, which is explained in the next subsection

*3.2.2 **Loss Function**.* We calculate Charbonnier loss [4] between restored image and ground truth to optimize the parameters of B-RSTB, the loss function $\mathcal{L}_c$ is formulated as Equation (2):

$$\mathcal{L}_c = \sqrt{||\hat{I}_{HQ} - I_{HQ}||^2 + \epsilon^2} \qquad (2)$$

where $\epsilon$ is a constant that is empirically set to $10^{-3}$.

Conducting progressive self-distillation in the early stage with only Charbonnier loss may compromise the upper-bound performance of deeper B-RSTBs. Inspired by prior works [38, 39], we introduce a channel-wise scoring distillation process to regulate parameter optimization, as depicted in Fig. 2 (c). A score map $\phi$ is computed through a learnable channel attention network and is defined as Equation (3):

$$\phi = \sigma(\Theta(AP(F)) + \Theta(MP(F))) \qquad (3)$$

where $\sigma$ denotes the sigmoid function, $F$ is the feature map and $F \in \mathcal{R}^{b \times c \times h \times w}$, $\Theta$ is a shared MLP, $AP$ and $MP$ represent average pooling and max pooing respectively. Then, the loss function $\mathcal{L}_{CA}$ is calculated as Equation (4), in which $F_t$ is feature map obtained from teacher model.

$$\mathcal{L}_{CA} = ||\sum_i^c F_{b,i,h,w} \cdot \phi_i - F_t||_1 \qquad (4)$$

To further leverage the hierarchical structure and ensure the best restoration quality, we minimize the total loss $\mathcal{L}$ through a weighted average following [17], indicated as Equation (5):

$$\mathcal{L} = \frac{\sum_{j=1}^n j \cdot (\alpha \cdot \mathcal{L}_{CA_j} + (1 - \alpha) \cdot \mathcal{L}_{c_j})}{\sum_{j=1}^n j} \qquad (5)$$

where $\alpha$ is a hyper-parameter which is empirically set to 1/10.

---

**Algorithm 1** Progressive Self-distillation

**Data**: $\mathcal{D} = \{I_{LQ_i}, I_{HQ_i}\}_i^N$
**Materials**: Max Epoch $E$, Schedule $S$, Teacher model $T$, Initialized model $M$, Loss $\mathcal{L}_{CA}$, $\mathcal{L}_c$ and Weight $\alpha$
**Result**: Self-distilled Model $M$

1: **Let** $e \longleftarrow 0$    // Initialize epoch
2: **Let** $P \longleftarrow 1$    // Initialize pointer
3: **while** $e < E$ **do**
4:    **Let** $d = S(e)$    // Get pointer index $d$ as scheduled
5:    **if** $P < d$ **then**
6:       **Activate** $P_{th}$ B-RSTB and $P + +$
7:    **end if**
8:    **Loop** $I_{LQ}, I_{HQ} = Next(\mathcal{D})$
9:    **for** $j = 1; j <= d; j + +$ **do**
10:      **if** $j == 1$ **then**
11:         $\hat{I}_{HQ_j}, F_j = $ B-RSTB$_j(I_{LQ_j})$ ; $F_t = T(I_{LQ})$
12:      **else**
13:         $\hat{I}_{HQ_j}, F_j = $ B-RSTB$_j(F_{j-1})$ ; $F_t = T(I_{LQ})$
14:      **end if**
15:      $\mathcal{L} = \alpha \cdot \mathcal{L}_{CA}(F_j, F_t) + (1 - \alpha) \cdot \mathcal{L}_c(\hat{I}_{HQ_j}, I_{HQ})$
16:      $M \longleftarrow Adam($B-RSTB$_j; \mathcal{L} \cdot j / \sum_{j=1}^d)$
17:   **end for**
18:   **End Loop IF** $\mathcal{D}$ is None
19:   **Do iteration** $e + +$
20: **end while**
21: **Return** $M$

---

## 3.3 Model Inference

In this section, we present two technical inference approaches: 1. Regular Branch Inference and 2. Adaptive Inference. After elucidating the mechanisms of these two approaches, we provide guidance on their application in diverse scenarios.

*3.3.1 **Regular Branch Inference**.* As previously introduced, a *switch* is associated with pointer to decide determine whether to engage the branch network. This allows us to activate partial FlexIR by controlling the pointer index, thereby achieving a diverse array of trade-offs between GPU cost, speed, and restoration quality within a single well-trained model. We observe that partitioning an image into regions (*e.p.* four regions) can effectively reduce computational costs with only a marginal performance degradation. thus we also seamlessly integrate it into our inference process. Further insights are provided in **Algorithm 2**.

*3.3.2 **Adaptive Inference**.* FlexIR is adept at performing inference adaptively, automatically activating certain B-RSTB modules and fast outputs based on the characteristics of the input image. The capability of Adaptive Inference is particularly beneficial when handling a wide variety of input categories, we will discuss it in next subsection.

One notable consensus is that, the difficulty of inference varies due to the inherent variations in image content. Therefore, a pivotal objective is to compute uncertainty scores for them. To this end, we introduce a dataset-irrelevant criteria for uncertainty estimation.

---

**Algorithm 2** Inference Approaches

**Input**: Image $I_{LQ}$, Pointer $P$, Threshold $h$

**Materials**: Partition operation $\mathcal{K}$, Union operation $\mathcal{U}$, Scoring operation $\Omega$

**Output**: Image $\hat{I}_{HQ}$

1: **Do** $\{I_1, I_2, ..., I_n\} = \mathcal{K}(I_{LQ})$     // Default: n=4
2: **Let** $i = 1$     // Initialize region index
3: **while** $i <= n$ **do**
4:    **if** Inference type == Regular **then**
5:       **for** $j = 1; j < P; j + +$ **do**
6:          $F_j = $ B-RSTB$_j(I_i)$   // *Switch* on
7:       **end for**
8:       **Let** $j == P$   // Output from last B-RSTB
9:       $\hat{I}_i = $ B-RSTB$_j(F_{j-1})$   // *Switch* off
10:    **else if** Inference type == Adaptive **then**
11:       **for** $j = 1; j <= P; j + +$ **do**
12:          $\hat{I}_i, F_j = $ B-RSTB$_j(I_i)$
13:          **if** $\Omega(\hat{I}_i) > h$ **then**
14:             Break and Fast Output $\hat{I}_i$
15:          **end if**
16:       **end for**
17:    **end if**
18:    **Do** iteration $i + +$
19: **end while**   // Then recover from regions
20: **Do** $\hat{I}_{HQ} = \mathcal{U}(\{I_1', I_2', ..., I_n'\})$
21: **Return** $\hat{I}_{HQ}$

---

We exploit the sharpness estimation method [48] in which maximum gradient and variability of gradients are utilized for scoring. To specify the calculation, a gradient map $G \in \mathcal{R}^{h \times w}$ is first generated by a gradient operator (*e.g.* Roberts operator), after clipping the gradient map to obtain the center of gradient map $G_c \in \mathcal{R}^{h-B \times w-B}$ and $B = round(min(h, w)/16)$, the maximum gradient is calculated as Equation (6):

$$MG = max(G_c) \qquad (6)$$

to capture content diversity from various regions, the gradient variability is computed as Equation (7):

$$VG = \frac{(max(G_c) - min(G_c))}{\sum_{i,j} G(i, j)/(h \times w)} \qquad (7)$$

Subsequently, the score $\Omega$ of an input image is obtained as Equation (8):

$$\Omega = MG^\beta \cdot VG^{1-\beta} \qquad (8)$$

where $\beta$ is a constant that is empirically set to 0.61 [48].

Since the score is exclusively derived from the image itself, it remains independent of the dataset and exhibits excellent generalization for uncertainty estimation. Moreover, due to the inherent variation in difficulty within individual parts of one image, partitioning images into regions aligns more suitably with Adaptive Inference.

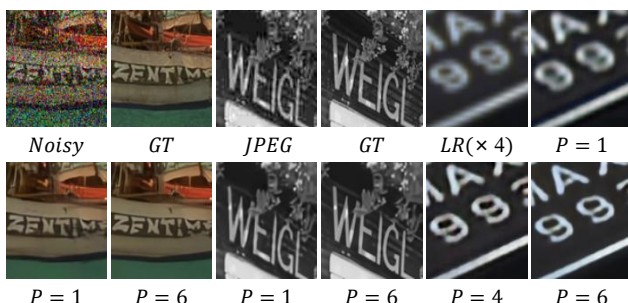

| Noisy | GT | JPEG | GT | LR(× 4) | P = 1 |
| P = 1 | P = 6 | P = 1 | P = 6 | P = 4 | P = 6 |

**Figure 3: Visual analysis reveals intriguing trends across different tasks (denoising, JPEG artifacts reduction, super-resolution). The setting of pointer should follow the real user requirements and the characteristic of executed task.**

## 3.4 Real-world Applications Analysis

In this section, we explore the practical applications of FlexIR in real-world scenarios, addressing questions such as the choice between Regular Branch Inference and Adaptive Inference, pointer settings, the advantages of these two inference approaches, and the determination of proper thresholds for Adaptive Inference.

### 3.4.1 Regular Branch Inference vs. Adaptive Inference

**Regular Branch Inference.** In scenarios where instances are homogeneous and share common characteristics, employing Regular Branch Inference, as elucidated earlier, is advisable. For instance, when dealing with similar cases (e.g., images of various fruits, images of various clothes), preemptive trade-offs can be identified.

**Adaptive Inference.** Conversely, in scenarios characterized by diversity [6], adopting Adaptive Inference is prudent, as strong priors are absent, and the nature of incoming cases is uncertain. Adaptive Inference allows specification of the desired quality of restored images by setting thresholds, with the scoring mechanism relying on inherent image features, ensuring robust generalization capabilities.

### 3.4.2 Deployment Strategy: Speed or Quality?

As a versatile instance adaptable for a myriad of scenarios, FlexIR can be flexibly manipulated according to specific user requirements in the inference stage, to determine speed-first or quality-first.

**The Setting of Pointer.** As depicted in Fig. 3, for tasks like image denoising or reducing JPEG compression artifacts, where visual differences are imperceptible, a smaller pointer number can be cost-effective while maintaining a fast response speed. However, for tasks like super-resolution, especially with larger scales, discernible differences emerge, necessitating a larger pointer number to ensure the quality of the outputs.

**The Setting of Threshold.** Threshold influences the efficiency when applying Adaptive Inference. Unfortunately, determining the optimal threshold is nearly unobtainable because we can not predict what kind of images user will upload into the model. Our empirical solution is to use user study methods (*e.p.* online AB test) with prepared a series of threshold values and adjust the threshold based on user feedback promptly.

Table 1: Quantitative results of color image denoising on benchmark datasets. The best and second-best results (PSNR) are colored by red and blue, respectively. FlexIR $P = 4$ indicates that four B-RSTBs are activated while all B-RSTBs are activated in *Full Size* FlexIR. $\sigma$ refers to the noise level, of which a larger value denotes a higher noise level.

| Method | | DnCNN [51] | IRCNN [52] | FFDNet [56] | DSNet [30] | BRDNet [34] | RNAN [54] | RDN [55] | IPT [5] | DRUNet [49] | CODE [57] | FlexIR [P = 4] | FlexIR [Full Size] |
|---|---|---|---|---|---|---|---|---|---|---|---|---|---|
| CSBD68 | $\sigma = 15$ | 33.90 | 33.86 | 33.87 | 33.91 | 34.10 | - | - | - | 34.30 | 34.33 | 34.39 | 34.39 |
| | $\sigma = 25$ | 31.24 | 31.16 | 31.21 | 31.28 | 31.43 | - | - | - | 31.69 | 31.69 | 31.75 | 31.75 |
| | $\sigma = 50$ | 27.95 | 27.86 | 27.96 | 28.05 | 28.16 | 28.27 | 28.31 | 28.39 | 28.51 | 28.47 | 28.52 | 28.53 |
| Kodak24 | $\sigma = 15$ | 34.60 | 34.69 | 34.63 | 34.63 | 34.88 | - | - | - | 35.31 | 35.32 | 35.32 | 35.32 |
| | $\sigma = 25$ | 32.14 | 32.18 | 32.13 | 32.16 | 32.41 | - | - | - | 32.89 | 32.88 | 32.87 | 32.87 |
| | $\sigma = 50$ | 28.95 | 28.93 | 28.98 | 29.05 | 29.22 | 29.58 | 29.66 | 29.64 | 29.86 | 29.82 | 29.76 | 29.77 |
| McMaster | $\sigma = 15$ | 33.45 | 34.58 | 34.66 | 34.67 | 35.08 | - | - | - | 35.40 | 35.38 | 35.58 | 35.59 |
| | $\sigma = 25$ | 31.52 | 32.18 | 32.35 | 32.40 | 32.75 | - | - | - | 33.14 | 33.11 | 33.28 | 33.28 |
| | $\sigma = 50$ | 28.62 | 28.91 | 29.18 | 29.28 | 29.52 | 29.72 | - | 29.98 | 30.08 | 30.03 | 30.16 | 30.16 |
| Urban100 | $\sigma = 15$ | 32.98 | 33.78 | 33.83 | - | 34.42 | - | - | - | 34.81 | - | 35.11 | 35.12 |
| | $\sigma = 25$ | 30.81 | 31.20 | 31.40 | - | 31.99 | - | - | - | 32.60 | - | 32.87 | 32.88 |
| | $\sigma = 50$ | 27.59 | 27.70 | 28.05 | - | 28.56 | 29.08 | 29.38 | 29.71 | 29.61 | - | 29.78 | 29.79 |

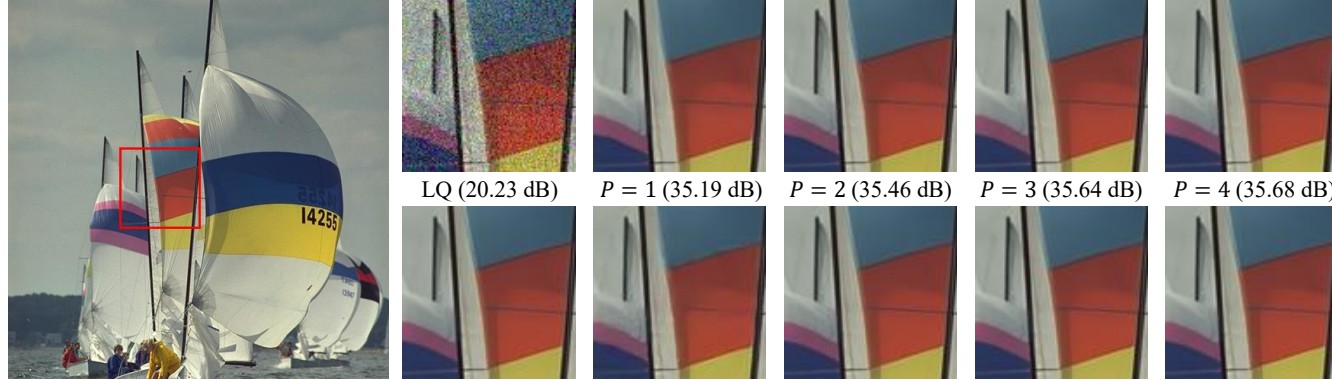

LQ (20.23 dB)    $P = 1$ (35.19 dB)    $P = 2$ (35.46 dB)    $P = 3$ (35.64 dB)    $P = 4$ (35.68 dB)

Ground Truth    $P = 5$ (35.68 dB)    DnCNN (34.59 dB)    Drunet (35.52 dB)    CODE (35.58 dB)    *Full Size* (35.68 dB)

Figure 4: Visual comparison of color image denoising (noise level 25) on image "kodim09" from Kodak24. Pointer $P$ indicates the number of activated B-RSTB while all B-RSTBs are activated in *Full Size* FlexIR.

Table 2: Quantitative results of JPEG compression artifact reduction on benchmark datasets. The best and second-best results (PSNR/SSIM/PSNRB) are colored by red and blue, respectively. FlexIR $P = 4$ indicates that four B-RSTBs are activated while all B-RSTBs are activated in *Full Size* FlexIR. $q$ refers to compression level, a smaller value denotes a higher compression level.

| Method | | DnCNN-3 [51] | RNAN [54] | RDN [55] | DRUNet [49] | CODE [57] | FlexIR [P = 4] | FlexIR [Full Size] |
|---|---|---|---|---|---|---|---|---|
| Classic5 | $q = 10$ | 29.40/0.8026/29.13 | 29.96/0.8178/29.62 | 30.00/0.8188/- | 30.16/0.8234/29.81 | 30.13/0.8225/- | 30.11/0.8219/29.81 | 30.25/0.8249/29.95 |
| | $q = 20$ | 31.63/0.8610/31.19 | 32.11/0.8693/31.57 | 32.15/0.8699/- | 32.39/0.8734/31.80 | 32.36/0.8731/- | 32.33/0.8702/31.80 | 32.50/0.8746/31.98 |
| | $q = 30$ | 32.91/0.8861/32.38 | 33.38/0.8924/32.68 | 33.43/0.8930/- | 33.59/0.8949/32.82 | 33.61/0.8951/- | 33.61/0.8948/32.94 | 33.72/0.8959/33.05 |
| | $q = 40$ | 33.77/0.9003/33.20 | 34.27/0.9061/33.4 | 34.27/0.9061/- | 34.41/0.9075/33.51 | 34.43/0.9078/- | 34.43/0.9073/33.65 | 34.52/0.9081/33.72 |
| LIVE1 | $q = 10$ | 29.19/0.8123/28.90 | 29.63/0.8239/29.25 | 29.67/0.8247/- | 29.79/0.8278/29.48 | 29.79/0.8281/- | 29.75/0.8262/29.39 | 29.84/0.8282/29.50 |
| | $q = 20$ | 31.59/0.8802/31.07 | 32.03/0.8877/31.44 | 32.07/0.8882/- | 32.17/0.8899/31.69 | 32.16/0.8901/- | 32.13/0.8891/31.59 | 32.23/0.8905/31.71 |
| | $q = 30$ | 32.98/0.9090/32.34 | 33.45/0.9149/32.71 | 33.51/0.9153/- | 33.59/0.9166/32.99 | 33.59/0.9168/- | 33.57/0.9160/32.88 | 33.66/0.9171/33.00 |
| | $q = 40$ | 33.96/0.9247/33.28 | 34.47/0.9299/33.66 | 34.51/0.9302/- | 34.58/0.9312/33.93 | 34.58/0.9313/- | 34.56/0.9307/33.80 | 34.65/0.9314/33.91 |

## 4  EXPERIMENT

### 4.1  Experiment Setup

**Implementation.** For all our experiments, we maintain uniform settings. Specifically, the B-RSTB number, STB number, window size, channel number and attention head number are generally set to 6, 6, 8, 180 and 6, respectively. B-RSTB consists of branch network and RSTB network, therefore we use RSTB parameters in SwinIR [21] to pre-train our FlexIR. **In this context, SwinIR can be regarded as a specific embodiment within the FlexIR**

**framework, distinguished primarily by its static model architecture**. All experiments are conducted in PyTorch framework with RTX 4090 GPU. For training, we use the Adam optimizer with $\beta_1 = 0.9$ and $\beta_2 = 0.999$. The learning rate is initialized to $1e^{-5}$, first increase then decrease through a linear warm-up strategy. For data augmentation, we use horizontal and vertical flips and obtain random $128 \times 128$ patches.

**Evaluation.** In our assessment of FlexIR's adaptability and user-directed manipulability, we undertake evaluations across three distinctive restoration tasks, **training a singular model for each while demonstrating various performance trade-offs through the adjustment of pointer**. Our evaluation involves two primary steps: initially, we benchmark the peak capabilities of the FlexIR model against various leading-edge methods by selecting a high pointer index. Subsequently, we detail FlexIR's performance metrics (#Params, MACs) at different pointer settings. This allows for a nuanced comparison with commonly employed models of equivalent scale or computational demand, including SwinIR, thereby illustrating FlexIR's efficiency and versatility in a comprehensive manner.

## 4.2 Color Image Denoising

For color image denoising, we train FlexIR on a composite dataset encompassing DIV2K [1], Flickr2k [35], BSD400 [2], and WED [27], we test the performance on CBSD68 [28], Kodak24 [13], McMaster [53] and Urban100 [15]. Consistent with existing methods [5, 30, 34, 49, 51, 52, 54–57], noise levels 15, 25 and 50 are used to test the PSNR performance on several benchmarks. Tab. 1 shows the quantitative results of color image denoising comparing with existing method while visual comparison is presented in Fig. 4.

## 4.3 JPEG Compression Artifact Reduction

To evaluate our method on JPEG compression artifact reduction, we train it on the same training datasets as color image denoising, and same with existing work [11, 49, 51, 54, 55, 57], we apply JPEG compression algorithm to images with quality factor of 10, 20, 30, 40 and test on two benchmark datasets: Classic5 [12] and LIVE1 [32]. Tab. 2 shows the comparisons of FlexIR with existing methods. visual comparison is presented in Fig. 5.

## 4.4 Real-world Image Super-Resolution

We also conduct experiments on Real-world image Super-Resolution, which is the ultimate goal of image SR for real-world applications. We test FlexIR on the real-world SR benchmark dataset RealSRSet [50]. In view of no ground-truth, we provide visual comparison with basic LR (×2 and ×4), and present visual images from FlexIR under different size, which is shown in Fig. 6. More visual results will will be presented in the supplementary material.

## 4.5 Analysis Experiments

In this section, we conduct experiments on individual components of FlexIR to better understand their effects. Our analysis consists of two aspects: model training and model inference.

**Analysis on Model Inference.** We present performance and cost comparisons of FlexIR under different settings against commonly used methods on McMaster (noise 25) and Classic5 (quality factor

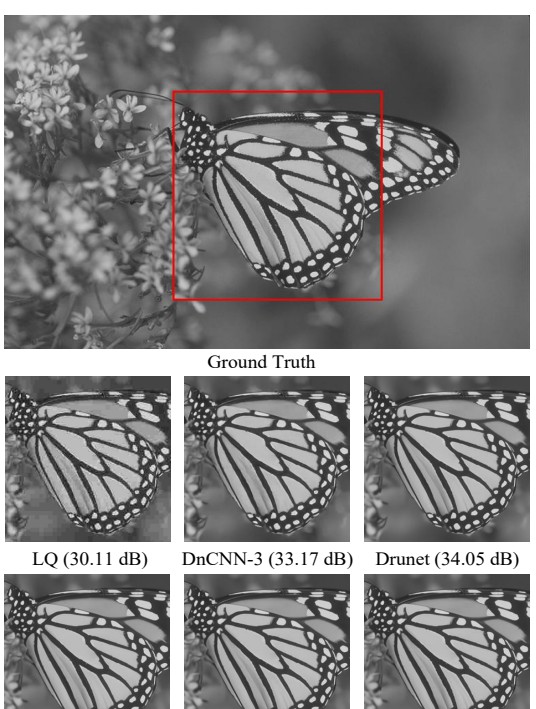

Ground Truth

| LQ (30.11 dB) | DnCNN-3 (33.17 dB) | Drunet (34.05 dB) |

| CODE (34.04 dB) | *P* = 4 (33.94 dB) | *Full Size* (34.12 dB) |

**Figure 5: Visual comparison of JPEG compression artifact reduction (quality factor 10) on image "monarch" from LIVE1. Pointer *P* indicates the number of activated B-RSTB while all B-RSTBs are activated in *Full Size* FlexIR.**

**Table 3: Comparison of Params, MACs, PSNR under different settings for color image denoising, *P* is the pointer number. PSNR is tested on McMaster (noise level 25), threshold for adaptive inference is 1.1×.**

| Method | #Params↓ | MACs↓ | PSNR↑ |
|---|---|---|---|
| **Full Size** | 11.46M | 188.03G | 33.28 |
| *P* = 5 | 9.60M | 157.52G | 33.28 |
| *P* = 4 | 7.74M | 127.01G | 33.28 |
| *P* = 3 | 5.88M | 96.50G | 33.25 |
| *P* = 2 | 4.02M | 65.98G | 33.12 |
| *P* = 1 | 2.16M | 35.47G | 32.81 |
| Adaptive* | 11.46M | - | 33.27 |
| **SwinIR [21]** | 11.46M | 188.03G | 33.20 |
| DnCNN [51] | 0.56M | 9.12G | 31.52 |
| DRUNet [49] | 32.64M | 35.90G | 33.14 |

40) in Tab. 3 and Tab. 4 respectively. The analysis reveals that while the full size FlexIR delivers superior performance, it also necessitates increased computational resources. Modifying the pointer to activate fewer B-RSTB layers results in notable savings in resource

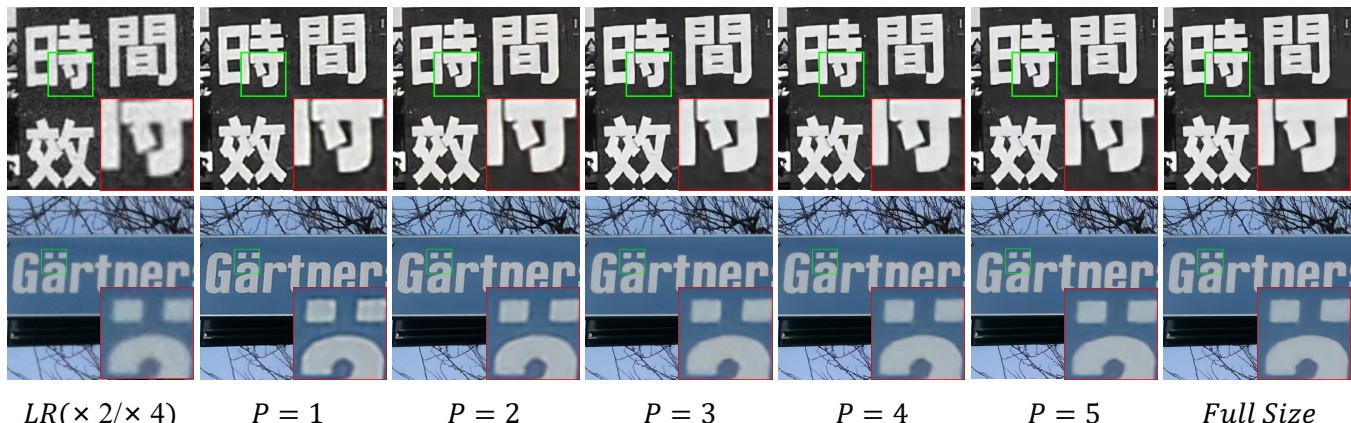

|     |     |     |     |     |     |     |
| :-: | :-: | :-: | :-: | :-: | :-: | :-: |
| *LR(× 2/× 4)* | *P = 1* | *P = 2* | *P = 3* | *P = 4* | *P = 5* | *Full Size* |

**Figure 6: Visual comparison of real-world image SR (×2: first row, ×4: second row) on RealSRSet. Compared images are derived from [50]. Pointer $P$ indicates the number of activated B-RSTB while all B-RSTBs are activated in "Full Size" FlexIR.**

**Table 4: Comparison of Params, MACs, PSNR under different settings for JPEG compression artifact reduction, $P$ is the pointer number. Performance (PSNR/SSIM/PSNRB) is tested on Classic5 (quality factor 40), threshold for adaptive inference is 1.1×. FlexIR surpass SwinIR on flexibility and upper-bound PSNRB performance.**

| Method | #Params↓ | MACs ↓ | Performance↑ |
| :-- | :-: | :-: | :-: |
| **Full Size** | 11.45M | 202.72G | 34.52/0.9081/33.72 |
| *P = 5* | 9.59M | 169.81G | 34.48/0.9078/33.70 |
| *P = 4* | 7.73M | 136.90G | 34.43/0.9073/33.65 |
| *P = 3* | 5.87M | 103.98G | 34.33/0.9064/33.52 |
| *P = 2* | 4.01M | 71.07G | 34.16/0.9045/33.42 |
| *P = 1* | 2.15M | 38.15G | 33.91/0.9013/33.41 |
| Adaptive* | 11.45M | - | 34.51/0.9081/33.70 |
| **SwinIR [21]** | 11.45M | 202.72G | 34.52/0.9082/33.66 |
| DRUNet [49] | 32.64M | 35.90G | 34.41/0.9075/33.51 |
| RDN [55] | 22.12M | 362.46G | 34.27/0.9061/- |
| RNAN [54] | 8.96M | 124.06G | 34.27/0.9061/33.40 |

consumption, though at a slight compromise in performance metrics. Despite this, FlexIR demonstrates competitive advantages in terms of PSNR/SSIM/PSNR-B, or exhibits lower memory requirements (#Params) when compared against analogous models, such as when setting $P = 3$ for FlexIR against RNAN [54] and DRUNet [49]. These comparative advantages are highlighted in the tables with purple and pink shading. In our adaptive inference assessment, we progressively adjust the inference threshold and evaluate the impact on PSNR/SSIM and average time cost, as shown in Tab. 5.

**Analysis on Model Training.** We experiment with training FlexIR while systematically abating the effects of Progressive Self-Distillation (PSD) and Channel-Wise Scoring distillation (C-WS). We follow a three-phase approach: initially training a pure FlexIR model using self-distillation (SD), then substituting SD with PSD, and finally incorporating the C-WS mechanism. As detailed in Tab. 6, by summing the average PSNR values obtained from all branches, we

observe an enhancement of **0.187 dB** as a result of employing both PSD and C-WS in the training process.

**Table 5: Performance and cost under different threshold. Increasing threshold lead to a more cautious decision-making by the model. PSNR/SSIM is computed on McMaster with noise level 25.**

| Threshold | PSNR↑ | SSIM↑ | Time Cost |
| :-: | :-: | :-: | :-: |
| 1.00× | 33.158 | 0.9023 | **14.25s** |
| 1.05× | 33.259 | 0.9044 | **21.18s** |
| 1.10× | 33.266 | 0.9045 | **22.34s** |
| 1.15× | 33.270 | 0.9046 | **22.46s** |

**Table 6: Ablation experiments for the components in FlexIR. PSNR is computed on McMaster with noise level 25.**

| Method | SD | PSD | C-WS | PSNR↑ |
| :-: | :-: | :-: | :-: | :-: |
| | ✓ | - | - | 32.982 |
| FlexIR | - | ✓ | - | 33.059 |
| | - | ✓ | ✓ | 33.169 (**+0.187**) |

## 5 CONCLUSION

In this work, we present FlexIR, a novel framework designed for image restoration, which integrates a hierarchical branch network, employs progressive self-distillation techniques, and utilizes channel-wise evaluation to achieve superior adaptability and efficiency. Distinctively, FlexIR empowers users to effectively manage the trade-offs between quality, cost, and efficiency, addressing the limitations inherent in existing models that depend excessively on auxiliary networks and are restricted by the biases present in their training datasets. Through rigorous experimentation, FlexIR has exhibited unparalleled flexibility and enhanced performance across a variety of conditions, signifying a considerable progression in fulfilling the complex demands of real-world applications. This framework not only advances the state-of-the-art in image restoration but also opens new avenues for user-centric model development.

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
