# OpenReview forum: "FlexIR: Towards Flexible and Manipulable Image Restoration"
_acmmm.org/ACMMM/2024/Conference — MM2024 Poster_

### Official Review · Reviewer_kfVn · 2024-05-24

**Rating:** 5
**Confidence:** 3

**Summary:**

This work introduces FlexIR, a framework designed to address the limitations of existing image restoration models, particularly in terms of training complexity and resource constraints. FlexIR features a hierarchical branch network, a progressive self-distillation process, and a channel-wise evaluation method to enhance flexibility and efficiency. It also proposes two inference methods, allowing users to balance quality, cost, and efficiency according to their needs. Experimental evaluations indicate that FlexIR performs well across various scenarios, demonstrating its adaptability and practical applicability in the field of image restoration.

**Strengths:**

This paper is well-written and successfully addresses potential research gaps in the image restoration. The authors present interesting ideas and provide detailed experimental analyses to support their contributions. FlexIR introduces a flexible and user-manipulable framework for image restoration, effectively addressing the limitations of existing models. At the core of FlexIR is a hierarchical network architecture enhanced by B-RSTB. This architecture enables dynamic output generation, allowing for real-time adjustments during inference without the need for retraining, which is a impressived trick over current methods that rely heavily on auxiliary networks and joint training. To enhance the training process, the authors propose a progressive self-distillation mechanism. This mechanism sequentially activates B-RSTBs using a pointer system, aiming to achieve optimal restoration quality incrementally. The introduction of a channel-wise scoring mechanism further refines the training process by effectively utilizing the features distilled from preceding blocks, improving overall efficiency and convergence.
For model inference, FlexIR incorporates two novel approaches: Regular Branch Inference and Adaptive Inference. The latter introduces a dataset-independent, gradient-based scoring method, making the framework highly adaptable to a variety of application scenarios. This dual inference methodology ensures that FlexIR can cater to diverse user needs and resource constraints, offering significant practical utility.
The experimental evaluations are favourable and demonstrate FlexIR's superiority. The fully-equipped FlexIR model performs on par with state-of-the-art alternatives, while its compact variant outperforms other small-scale methods. Importantly, the framework's manually adjustable operational parameters during inference eliminate the need for complex retraining, underscoring its versatility and cost-effectiveness.

**Limitations:**

While the paper excels in problem description, method introduction, result presentation, and experimental design and analysis, further empirical analysis on different hardware devices and hyperparameters would strengthen the study.

**Suitability:**

3

---

### Official Review · Reviewer_ovAy · 2024-05-24

**Rating:** 6
**Confidence:** 4

**Summary:**

The paper introduces FlexIR, a novel framework designed for image restoration that offers dynamic and manipulable features. Unlike traditional models that are either static or dependent on auxiliary networks for dynamic processing, FlexIR employs a hierarchical branch network, progressive self-distillation, and channel-wise evaluation methods. These innovations allow FlexIR to adapt to different user requirements and deployment contexts without the need for retraining. Experimental results demonstrate FlexIR's superior flexibility and performance across various image restoration tasks.

**Strengths:**

The innovative training mechanisms of proposed method, including progressive self-distillation and channel-wise scoring distillation, significantly enhance training efficiency and effectiveness, leading to superior performance across multiple image restoration tasks such as color image denoising and JPEG compression artifact reduction. Additionally, the user-controlled inference capability provides practical value by enabling users to balance quality, cost, and efficiency based on specific requirements.

**Limitations:**

Incorporating validation experiments on different hardware devices would more effectively demonstrate the superior performance of the proposed method.

**Suitability:**

3

---

### Official Review · Reviewer_6SHL · 2024-05-26

**Rating:** 2
**Confidence:** 4

**Summary:**

This paper introduces FlexIR: a flexible and actionable framework for image restoration.FlexIR consists of three components: a hierarchical branching network, progressive self distillation, and a channel evaluation method for improving the efficiency of knowledge distillation. Experimental evaluations validate the effectiveness of FlexIR.

**Strengths:**

The proposed method is valuable for real-world applications of image restoration. The formulation of the method is clear.

**Limitations:**

1.Overall, the writing of the paper needs to be strengthened. The authors do not clearly present the motivation for this approach. For example the rationale for estimating importance based on gradient maps.

2.The proposed method does not have any efficiency advantage over other methods. For example, in Table 1, the competing method CODE requires only 22.52G FLOPs while the proposed FiexIR-4 requires 136G FLOPs.

3.Moreover, the qualitative experimental results do not show the advantages of the proposed method.

**Suitability:**

2

---

### Official Review · Reviewer_R9hJ · 2024-05-26

**Rating:** 2
**Confidence:** 4

**Summary:**

This paper presents a flexible method for image restoration. A channel-wsie self-distillation technique is introduced. Experimental results demonstrate the effectiveness of the method.

**Strengths:**

Overall, the proposed method is completely presented. The whole paper is well structured with extensive and comprehensive experiments.

**Limitations:**

1. The proposed method is highly dependent on empirical parameter settings. Such as alpha in Eq. 5 and beta in Eq. 8. The authors need to analyze these hyperparameters.

2. The authors claim to have proposed the method to solve the problem of model deployment for real applications. However, the most critical inference time is not included in the Table 1 and 2.

3. Comparison of real world image SR lacks comparison algorithms. Why not do it in bicubic setting so that you can compare some similar techniques like ClassSR[1] and CAMixerSR[2].

4. FlexIR (fullsize) seems to perform weaker than SwinIR with the same parameters and FLOPs. Maybe the author can give an explanation.

> 1.Classsr: A general framework to accelerate super-resolution networks by data characteristic.CVPR 2021.

> 2.CAMixerSR: Only Details Need More" Attention". CVPR 2024.

**Suitability:**

2

---

### Meta-Review · Area_Chair_1vYB · 2024-07-03

**Recommendation:** Accept (Poster)
**Confidence:** 4

**Metareview:**

The paper introduces FlexIR, a novel framework designed for image restoration that offers dynamic and manipulable features. This paper addresses a highly practical application scenario. The authors present interesting ideas and provide detailed experimental analyses to support their contributions. Overall, the paper meets the acceptance bar. However, the authors still need to address the reviewers' concerns in the final version, including improving the writing and providing a comparison of real-world image super-resolution.